# Self-Discover: Large Language Models Self-Compose Reasoning Structures

**Pei Zhou**◇    **Jay Pujara**◇    **Xiang Ren** ◇    **Xinyun Chen**†    **Heng-Tze Cheng**†
**Quoc V. Le**†    **Ed H. Chi**†    **Denny Zhou**†    **Swaroop Mishra**†    **Huaixiu Steven Zheng**†
† Google DeepMind    ◇ University of Southern California

## Abstract

We introduce SELF-DISCOVER, a general framework for LLMs to self-discover the task-intrinsic reasoning structures to tackle complex reasoning problems that are challenging for typical prompting methods. Core to the framework is a self-discovery process where LLMs select multiple atomic reasoning modules such as critical thinking and step-by-step thinking, and compose them into an explicit reasoning structure for LLMs to follow during decoding. SELF-DISCOVER substantially improves GPT-4 and PaLM 2's performance on challenging reasoning benchmarks such as BigBench-Hard, grounded agent reasoning, and MATH, by as much as 32% compared to Chain of Thought (CoT). Furthermore, SELF-DISCOVER outperforms inference-intensive methods such as CoT-Self-Consistency by more than 20%, while requiring 10-40x fewer inference compute. Finally, we show that the self-discovered reasoning structures are universally applicable across model families: from PaLM 2-L to GPT-4, and from GPT-4 to Llama2, and share commonalities with human reasoning patterns.

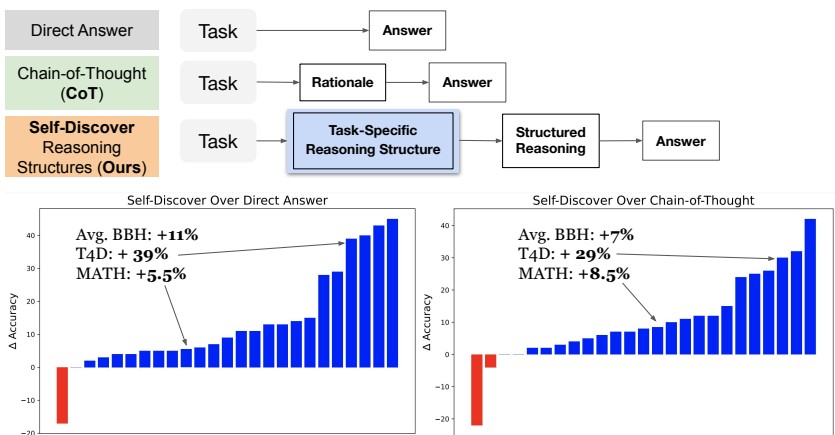

Figure 1: SELF-DISCOVER guides LLMs to self-discover and compose atomic reasoning modules into a reasoning structure to solve challenging tasks. Through testing on challenging reasoning benchmarks incuding Big Bench-Hard (BBH), agent reasoning (T4D), and MATH, we find that SELF-DISCOVER outperforms Direct Answering on 23/25 and CoT on 21/25 tasks in zero-shot setting using PaLM 2-L. Full BBH results are in Appendix C Table 3.

## 1   Introduction

Large Language Models (LLM) (Brown et al., 2020; Chowdhery et al., 2022; OpenAI, 2023b; Anil et al., 2023) powered by transformers (Vaswani et al., 2017) have produced impressive breakthroughs

38th Conference on Neural Information Processing Systems (NeurIPS 2024).

in generating coherent texts (OpenAI, 2022), and following instructions (Zhong et al., 2021; Mishra et al., 2022c; Wei et al., 2021; Chung et al., 2022; Ouyang et al., 2022). In pursuit of the goal to enhance LLMs' capability to *reason* and solve complex problems, various prompting methods have been proposed, drawing inspirations from cognitive theories of how humans reason. For example, few-shot and zero-shot chain-of-thought (CoT) (Nye et al., 2021; Wei et al., 2022; Kojima et al., 2022; Yasunaga et al., 2023) resembles how humans solve problems step-by-step, decomposition-based prompting (Zhou et al., 2022a; Drozdov et al., 2022; Patel et al., 2022; Hao et al., 2023; Khot et al., 2022) is inspired by how humans breakdown a complex problem into a series of smaller subproblems, and then solve those subproblems one by one (Polya, 2004), and step-back prompting (Zheng et al., 2023) is motivated by how humans reflect on task nature to derive general principles. However, a fundamental limitation is that each technique itself serves as an atomic reasoning module making an implicit prior assumption of the process on how to tackle a given task. Instead, we argue that each task has a unique intrinsic structure underlying the reasoning process involved in solving it efficiently. For instance, least-to-most prompting (Zhou et al., 2022a; Drozdov et al., 2022) has shown to be much more effective than CoT (Wei et al., 2022) at solving tasks such as symbolic manipulation and compositional generalization, due to the decomposition structure of the tasks.

This paper aims at self-discovering the underlying reasoning structure unique to each task, while being highly efficient in terms of computation. Our approach, SELF-DISCOVER, is inspired by how humans internally devise a reasoning *program* for problem-solving (Newell et al., 1958; Rasmussen, 1983), as illustrated in Figure 2 . From a set of atomic reasoning modules described in natural language such as "*breakdown into sub tasks*" and "*critical thinking*", an LLM, and task examples without labels, SELF-DISCOVER composes a coherent reasoning structure intrinsic to the task (Stage 1) and then solves instances of the task using the discovered structure (Stage 2). Stage 1 operates at the task-level and uses three actions to guide the LLM to generate a reasoning structure for the task. At Stage 2, during the final decoding, the LLM simply follows the self-discovered structure to arrive at the final answer.

Solving problems using SELF-DISCOVER brings several benefits compared to other methods for LLM reasoning. First, the discovered reasoning structure is grounded in atomic reasoning modules benefiting from the strengths of *multiple* reasoning modules in contrast to applying a priori module such as CoT. Second, SELF-DISCOVER is efficient in computation as it only requires 3 more inference steps on the *task-level*, while being more performant than inference-heavy ensemble approaches such as self-consistency Wang et al. (2022). Lastly, the discovered reasoning structure is intrinsic to the task, and conveys LLMs' insights about the task in a more *interpretable* way than the optimized prompts (Zhou et al., 2022b; Yang et al., 2023).

We test SELF-DISCOVER on 25 challenging reasoning tasks including Big Bench-Hard (BBH) (Suzgun et al., 2022), Thinking for Doing (T4D) (Zhou et al., 2023) and MATH (Hendrycks et al., 2021). SELF-DISCOVER outperforms CoT on 21/25 task with performance gains up to 42% (Figure 1), highlighting the advantage of the self-discovered reasoning structure composed from the atomic reasoning modules against a single a priori CoT module. Furthermore, we demonstrate that SELF-DISCOVER achieves superior performance against inference-heavy methods such as CoT + Self-Consistency and majority voting of every module while requiring 10-40x fewer inference compute (Figure 5). Finally, we compare SELF-DISCOVER with prompts optimized (OPRO) using a training set (Yang et al., 2023) (Figure 8). We find that SELF-DISCOVER still performs on par or better than OPRO while the self-discovered reasoning structure are much more interpretable.

We conduct a set of analysis to understand the effectiveness of SELF-DISCOVER. By breaking down BBH tasks into 4 different categories, we find that SELF-DISCOVER performs best on tasks requiring world knowledge and has a moderate performance boost on algorithmic tasks compared to CoT (Figure 4). This is further confirmed by the error analysis on MATH, where 74.7% model failures comes from computation errors (e.g. math). We also take a closer look at the self-discovered reasoning structures, and show the universality of them by transferability study from PaLM 2-L to GPT-4, and from GPT-4 to Llama-2-70B. We hope to encourage more future work on structured reasoning for solving challenging problems using LLMs.

## 2   Self-Discovering Reasoning Structures for Problem-Solving

We take inspiration from how humans use prior knowledge and skills to devise a reasoning program to solve problems (Newell et al., 1958; Rasmussen, 1983). When we face a new problem, we often first search internally what knowledge and skills from our prior experience might be helpful to solve it. Then we will attempt to apply relevant knowledge and skills to this task. And finally we will connect multiple individual skills and knowledge to solve the problem. We design SELF-DISCOVER to enact these steps into two stages as illustrated in Figure 2.

Given a task and a set of reasoning module descriptions representing high-level problem-solving heuristics such as "*Use critical thinking*" and "*Let's think step by step*", Stage 1 of SELF-DISCOVER aims to uncover the intrinsic reasoning structure for solving this task via meta-reasoning. Specifically, we uses three meta-prompts to guide LLMs to select, adapt, and implement an actionable reasoning structure with no labels or training required. We format the structure in key-value pairs similar to JSON due to interpretability and findings on following JSON boosts reasoning and generation quality (Zhou et al., 2023; OpenAI, 2023a). The structure of the meta-prompts and full prompts are shown in Figure 9. Stage 1 operates on *task-level*, meaning we only need to run SELF-DISCOVER once for each task. Then, in Stage 2, we can simply use the discovered reasoning structure to solve every *instance* of the given task by instructing models to follow the provided structure by filling each key and arrive at a final answer.

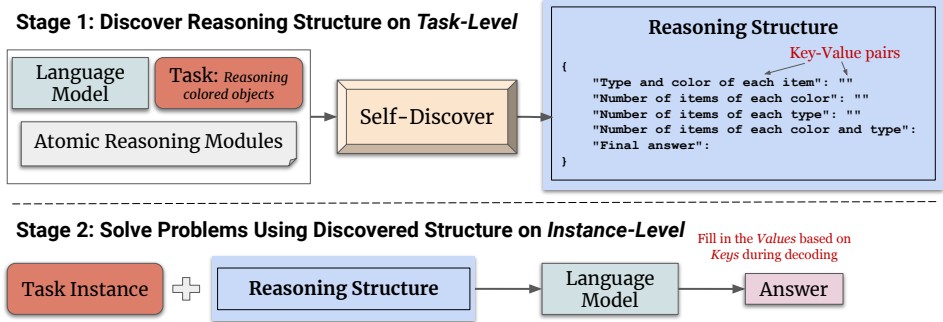

Figure 2: **Illustration of using SELF-DISCOVER for problem-solving**. Given a generative LM, task, and seed reasoning module descriptions, we guide LMs to generate a reasoning structure in *key-value* format to solve the task. Finally, models can follow the self-discovered structures to solve the every instance from the task by filling in the values in JSON step-by-step.

### 2.1   Stage 1: Self-Discover Task-Specific Structures

The first stage consists of three actions: 1) SELECT, where relevant reasoning modules for task-solving are chosen from the set of reasoning module descriptions; 2) ADAPT, where descriptions of selected reasoning modules are rephrased to be more specific to the task at hand; and 3) IMPLEMENT, where the adapted reasoning descriptions are implemented into a structured actionable plan so that the task can be solved by following the structure.

**SELECT**   First, not every reasoning module is helpful for every task, so the first stage of SELF-DISCOVER guides model to select modules that are useful based on task examples. For example, "*reflective thinking*" might help search for first-principle theories on science problems, while "*creative thinking*" helps on generating a novel continuation to a story. Given raw set of reasoning module descriptions $D$ such as "*critical thinking*", and "*break the problem into sub-problems*" (full set in Appendix A), and a few task examples without labels $t_i \in T$, SELF-DISCOVER first selects a subset of reasoning modules $D_S$ that are useful for solving the tasks by using a model $\mathcal{M}$ and a meta-prompt $p_S$:

$$D_S = \mathcal{M}(p_S \parallel D \parallel t_i). \qquad (1)$$

**ADAPT**   Since each reasoning module provides a general description of how to solve problems, the next step of SELF-DISCOVER aims at *tailoring* each selected module to the task at hand. For example, from "*break the problem into sub-problems*" to "*calculate each arithmetic operation in order*" for arithmetic problems. Given selected reasoning module subset $D_S$ from the previous step, ADAPT

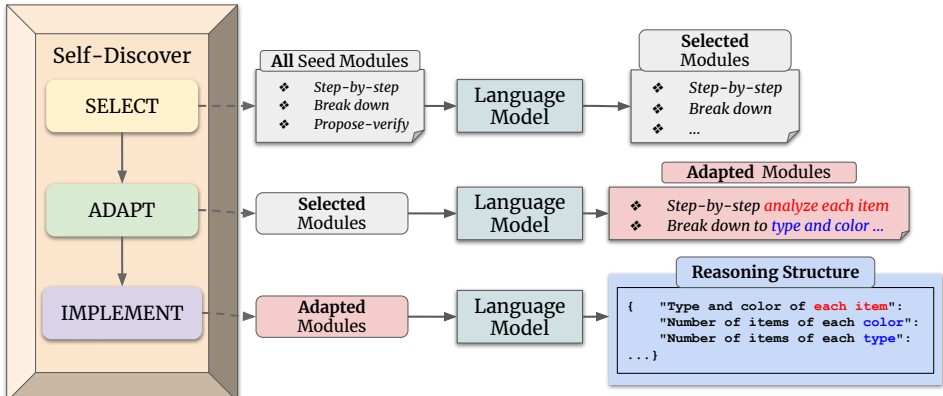

Figure 3: **Illustration of three actions of SELF-DISCOVER**. We use LMs to compose a coherent reasoning structure by selecting relevant modules, adapting to task-specific descriptions, and implement a reasoning structure in JSON.

rephrases *each* of the selected module to be more specific to the task. Similarly to SELECT, this stage uses a meta-prompt $p_A$ and a generative model $\mathcal{M}$ to generate the adapted reasoning module descriptions $D_A$:

$$D_A = \mathcal{M}(p_A \parallel D_S \parallel t_i). \tag{2}$$

**IMPLEMENT** Finally, given the adapted reasoning module descriptions $D_A$, SELF-DISCOVER operationalizes the reasoning modules into an implemented *reasoning structure* $D_I$ with specified instruction on what to generate for each step. In addition to a meta prompt $p_I$, IMPLEMENT also provides a demonstration of a human-written reasoning structure $S_{human}$ on another task to better convert the natural language descriptions into a reasoning structure:

$$D_I = \mathcal{M}(p_I \parallel S_{human} \parallel D_A \parallel t_i). \tag{3}$$

## 2.2 Stage 2: Tackle Tasks Using Discovered Structures

After the three stages, we have an implemented reasoning structure $D_I$ uniquely adapted for the task we need to solve $T$. Then we can simply append the reasoning structure to all instances of the task and prompt models to follow the reasoning structure to generate an answer $A$:

$$A = \mathcal{M}(D_I \parallel t), \forall t \in T. \tag{4}$$

## 3 Experiment Setup

### 3.1 Tasks

We focus on diverse reasoning benchmarks that are still challenging for LLMs: BIG-Bench Hard (BBH) (Suzgun et al., 2022) contains 23 carefully-selected challenging tasks from BIG-Bench (Srivastava et al., 2023). BBH tasks cover a diverse range of reasoning problems spanning the following 4 categories according to their authors: 1) Algorithmic and Multi-Step Arithmetic Reasoning, 2) Natural Language Understanding, 3) Use of World Knowledge, and 4) Multilingual Knowledge and Reasoning. We also test on a grounded social agent reasoning task called Thinking for Doing (T4D) where models must leverage mental state reasoning to determine actions to perform (Zhou et al., 2023), where GPT-4 with CoT only reaches around 50%. Finally, we subsample 200 examples from the MATH (Hendrycks et al., 2021) test set, and generate instance-level reasoning structures via a one-shot demonstration to adapt to the complexity of MATH tasks. For evaluations, we use accuracy to measure the model performance on BBH, T4D and MATH (details can be found in Appendix B).

### 3.2 Models

We use several state-of-the-art LLMs: GPT-4 (gpt-4-turbo-preview) (OpenAI, 2023b), GPT-3.5-turbo (ChatGPT) (OpenAI, 2022)[1], instruction-tuned PaLM 2-L (Anil et al., 2023)[2], and an open-source LLM Llama2-70B (Touvron et al., 2023).

### 3.3 Baselines

We compare SELF-DISCOVER with other zero-shot prompting methods for LLM reasoning:

- **Direct Prompting**, where model directly generates the answer without intermediate reasoning steps.
- **CoT** (Wei et al., 2022; Kojima et al., 2022), where models are prompted to generate a reasoning process leading to the final answer.
- **Plan-and-Solve** (Wang et al., 2023), where models are prompted to first generate a plan and then solve the problem. SELF-DISCOVER differs by grounding the reasoning structure in atomic reasoning modules, and prompting the decoding to follow the explicit key-value reasoning structure.

Next, we also consider other baselines that make use of the raw seed reasoning modules (RM) we pass to SELF-DISCOVER. We compare with the following methods' performance and the inference call efficiency on a subset of tasks.

- **CoT-Self-Consistency** Wang et al. (2022), we sample multiple outputs from LLM with CoT and aggregate answers to get the final answer. We compare this method on a subset of tasks due to the cost of repetitive queries.
- **Majority voting of each RM**: we prompt models to solve the tasks by appending each RM and use majority voting of all answers to get the final answer. We examine whether integrating *multiple* RMs into a coherent reasoning structure is advantageous to applying each RM to solve the task and use majority voting to ensemble them post-hoc, which costs much more inference computation.
- **Best of each RM**: this method assumes that we have access to oracle labels and uses the highest accuracy from applying each RM. We compare with this to examine whether SELF-DISCOVER competes with methods that depend on perfect prior knowledge of which RM to use on a new task.

Furthermore, for analysis on universality of reasoning structures, we compare with a prompt-optimization method that require a *training* set to improve prompts: LLMs as optimizers (**OPRO**) (Yang et al., 2023). We aim to show that when we apply structures or prompts optimized from one model, the reasoning structures can retain more performance gains than the wordings of prompts.

## 4 Results

We answer the following questions through experimental results: 1) *Does discovering reasoning structures improve LLM reasoning capabilities?* (4.1) 2) *Which categories of problems do* SELF-DISCOVER *perform the best?* (4.2) and 3) *Can* SELF-DISCOVER *boost LLM performance efficiently?* (4.3) Finally, we will show qualitative examples of self-discovered structures, LLM output following the structures, and compare with LLM output following other prompting methods for reasoning (4.4).

### 4.1 Does SELF-DISCOVER Improve LLM Reasoning?

**Overall, SELF-DISCOVER improves PaLM 2-L and GPT-4's reasoning across diverse set of reasoning tasks.** Table 1 shows the overall results on complex reasoning tasks of BBH, T4D and

---

[1]accessed in October-December 2023

[2]For MATH, we use a PaLM 2-L model with a stronger instruction tuning to enable better instruction following of more complex reasoning structures.

Table 1: Self-Discover significantly improves LLM reasoning across a diverse set of 25 complex tasks: BBH, T4D and MATH. CoT: zero-shot Chain of Thought (Kojima et al., 2022). PS: plan-and-solve prompting (Wang et al., 2023).

| Method | BBH | T4D | MATH |
|---|---|---|---|
| PaLM 2-L | 56% | 30% | 45% |
| PaLM 2-L + CoT | 60% | 40% | 42% |
| PaLM 2-L + PS | 61% | 42% | 49% |
| PaLM 2-L + Self-Discover | **67%** | **69%** | **50.5%** |
| GPT-4 | 58% | 51% | 70.5% |
| GPT-4 + CoT | 75% | 52% | 71% |
| GPT-4 + PS | 73% | 53% | 70% |
| GPT-4 + Self-Discover | **81%** | **85%** | **73%** |

MATH using PaLM 2-L and GPT-4. We compare Self-Discover with baselines including direct prompting, CoT, and Plan-and-Solve (PS).

On aggregated 23 tasks of BBH, SELF-DISCOVER achieves 7% and 6% absolute improvement on PaLM 2-L over Chain-of-Thought and Plan-and-Solve, respectively. Similar gains (6% and 8%) are observed when SELF-DISCOVER is applied to GPT-4. Breakdown results of each task's improvement over direct answering and CoT of PaLM 2-L are shown in Figure 1, where we find SELF-DISCOVER outperforms them on over 20/24 tasks. For a per-task performance for all 23 BBH tasks, please refer to Appendix C.

On the grounded social agent task T4D, SELF-DISCOVER reaches over $\geq 27\%$ (32%) absolute improvement over all baselines on PaLM 2-L (GPT-4). SELF-DISCOVER achieves 69% and 85% accuracy on PaLM 2-L and GPT-4, significantly outperforming previous SoTA prompting method such as Foresee and Reflect (FaR) which employs an expert-designed reasoning structure. In contrast, SELF-DISCOVER generates the reasoning structure automatically from a set of atomic reasoning modules without human interventions.

For MATH, we observe a moderate gain of 1%-7% (2%-3%) on PaLM 2-L (GPT-4) from SELF-DISCOVER compared to the baselines. Upon error analysis (see Appendix E for details), we find that the reasoning structures generated by PaLM 2-L from SELF-DISCOVER are correct 87.5% of the time: human experts can follow the reasoning structures to solve the tasks perfectly. The majority of the failures (74.7%) comes from errors in executing the computations, consistent with prior findings Zheng et al. (2023).

## 4.2 Which Types of Problems Do SELF-DISCOVER Help the Most?

**SELF-DISCOVER performs best on tasks that require *diverse world knowledge*.** Figure 4 presents the average improvement in terms of delta in accuracy of SELF-DISCOVER over direct answer and CoT on 4 categories of reasoning tasks we test. We adopt the categorization from Suzgun et al. (2022). We find that SELF-DISCOVER improves over these two baselines on all categories, but especially on tasks that require world knowledge such as *sports understanding*, *movie recommendation*, and *ruin names*.

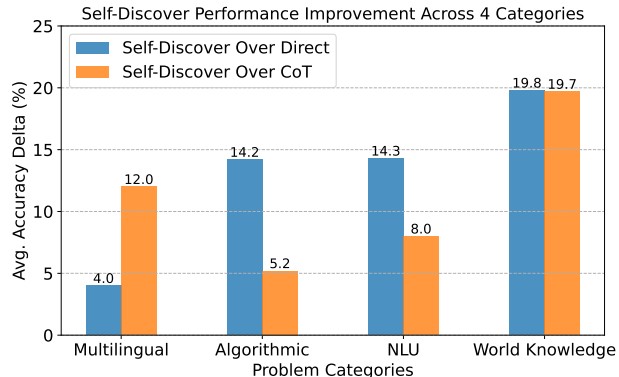

Figure 4: **Breakdown of SELF-DISCOVER performance improvement on 4 categories on PaLM 2-L**. SELF-DISCOVER performs the best on tasks requiring world knowledge.

These tasks demand models to reason using fact and general commonsense knowledge. We interpret SELF-DISCOVER's advantages on these tasks as strength from integrating multiple reasoning modules from various perspectives as only applying CoT might miss key knowl-

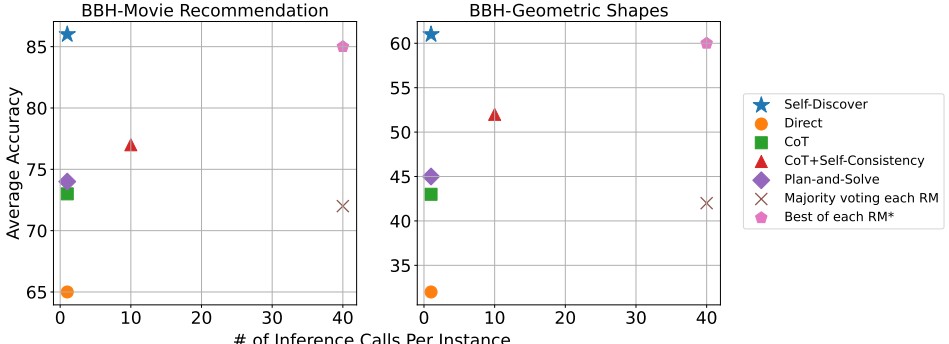

Figure 5: **Comparison of accuracy with number of inference calls required per instance**. For CoT-Self-Consistency, we sample 10 times. Best of each RM method requires gold labels (*). SELF-DISCOVER requires only 1 inference call per instance, same as Direct and CoT while reaching better performance compared with 40x more call required methods on GPT-4. We acknowledge that SELF-DISCOVER input and output are longer than CoT and Direct prompting, increasing cost. However, as the number of instances increases, the efficiency of SELF-DISCOVER in terms of inference per instance is highly desirable.

edge in the reasoning process. We observe that the gain on the Algorithmic category is moderate, consistent with the findings from Sec. 4.1 on MATH.

### 4.3 How Efficient is SELF-DISCOVER?

**SELF-DISCOVER achieves better performance while requiring 10-40x fewer inference computer compared to self-consistency or majority voting.** Here we examine a subset of 2 tasks from BBH and present a more thorough comparison of methods including those requiring many inference calls that are too costly to run on all 24 tasks. Figure 5 shows average accuracy and number of inference calls required per instance for each method using GPT-4. **Accuracy wise (y-axis)**, we find that SELF-DISCOVER outperforms other baselines even those that require repeated inference calls such as CoT-self-consistency and majority voting of applying each RM. **Efficiency wise (x-axis)**, SELF-DISCOVER only requires one call per instance and three more inference calls on the task-level, CoT-self-consistency requires 10 times more since we have to sample 10 times for each instance, and methods using each RM requires 40 times more as we use 40 RMs. In summary, SELF-DISCOVER presents itself a strong reasoning boosting method that is efficient to deploy on large-scale.

### 4.4 Qualitative Examples

We show examples of model-discovered structures for different reasoning tasks in Figure 10 from PaLM 2-L. We observe that each structure is uniquely adapted to the task, integrates multiple reasoning modules, and provides insights on how to solve the tasks. Furthermore, example of comparing reasoning processes from CoT, Plan-and-Solve, and SELF-DISCOVER is shown in Figure 6. We find that CoT and Plan-and-Solve makes incorrect assertions early and arrives at a wrong answer while following structure from SELF-DISCOVER leads the model to generate logical conclusions ("*path is closed as the beginning and ending coordinates are the same*") and arrive at the correct answer.

## 5 Deep Diving Into Self-Discovered Reasoning Structures

After experimental results showing the effectiveness and efficiency of SELF-DISCOVER on a range of reasoning tasks, this section further analyzes **are all actions of SELF-DISCOVER needed** and **what other benefits can self-discovered structures bring?** In Sec. 5.1, we show that it is critical to the model's performance to use the reasoning structures discovered through the three steps of SELECT, ADAPT and IMPLEMENT. In Sec. 5.2, we demonstrate the **universality** of the self-discovered reasoning structures by (1) applying the structures discovered by PaLM 2-L to GPT-4, (2) applying the structures discovered by GPT-4 to Llama-2-70B. We further show the commonalities between the reasoning structures and human reasoning patterns in Appendix F.

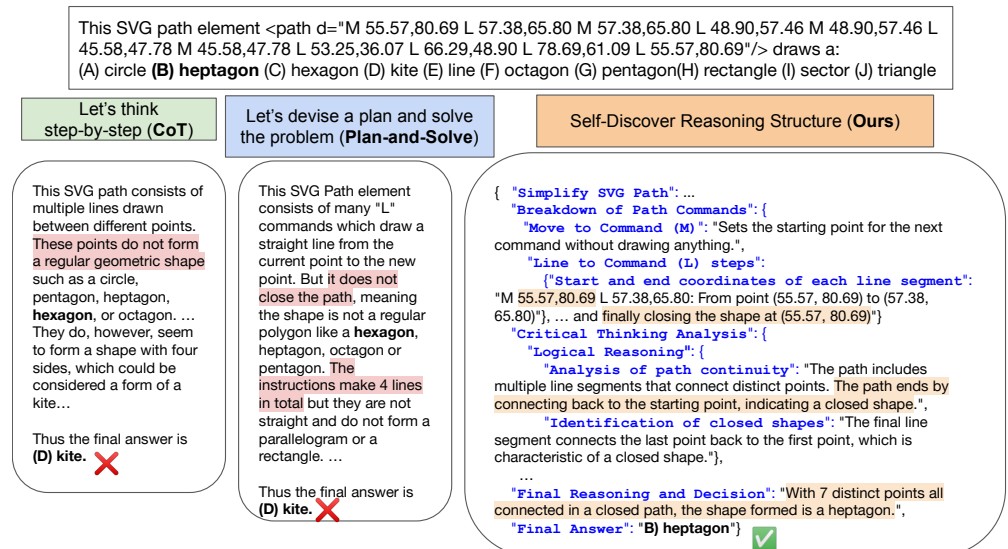

Figure 6: **Comparison of generated reasoning process from CoT, Plan-and-Solve, and SELF-DISCOVER** on BBH-geometric shape task. Both CoT and Plan-and-Solve incorrectly asserts that the path does not form a regular shape as it is not a closed path (highlighted in red) and arrive at a wrong answer. The reasoning structure (in **blue Courier font**) from SELF-DISCOVER first breaks down each line segment and analyze the coordinates carefully, then leverages logical reasoning to conclude that it forms a closed shape as the path ends at the same coordinate (highlighted in purple and orange), and selects the correct answer through final reasoning.

## 5.1 Importance of SELF-DISCOVER Actions

We conduct ablation study on the three actions: SELECT, ADAPT, and IMPLEMENT to analyze the effects of SELF-DISCOVER actions. Figure 7 show results using GPT-4 on 4 reasoning tasks when we apply SELECT (-S) or apply SELECT and ADAPT (-SA) or apply all three actions. We find that with each stage, model's zero-shot reasoning capability improve consistently across tasks, indicating that all three actions are beneficial.

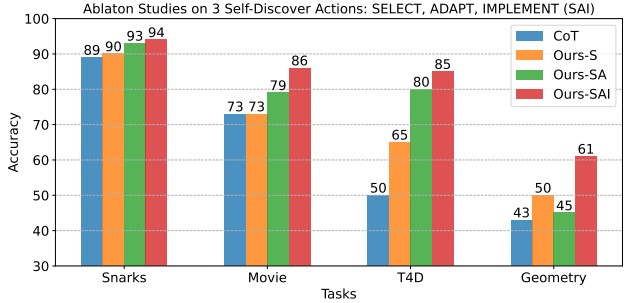

Figure 7: **Ablation study on three SELF-DISCOVER actions on 4 reasoning tasks**: all three actions are beneficial for task-solving.

## 5.2 Towards Universality of Discovered Reasoning Structures

**Applying PaLM 2-L Discovered Structures to GPT-4** We first use a PaLM 2-L model to discover the reasoning structures of 4 reasoning tasks. Then, we apply the resulting reasoning structures to the decoding of GPT-4 as grounding. We compare our approach to OPRO (Yang et al., 2023) which discovered zero-shot-prompts through optimizations. We apply OPRO prompts optimized using PaLM 2-L on each task to GPT-4 on the same reasoning tasks. Figure 8 shows that SELF-DISCOVER outperforms OPRO on 3 out of 4 tasks despite that OPRO used 20% data to op-

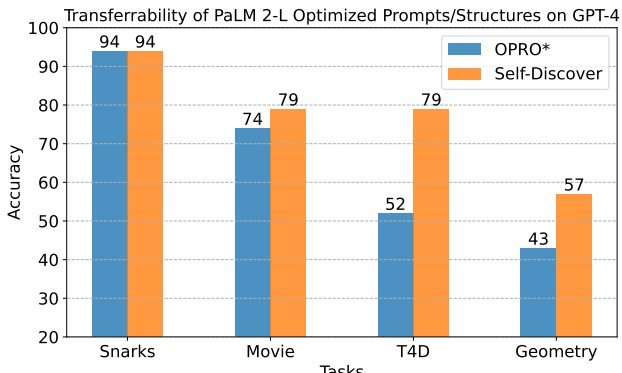

Figure 8: **Transferrability tests of optimized prompts (OPRO) and composed structures (SELF-DISCOVER).**

timize the prompt. In contrast, SELF-DISCOVER is done in a zero-shot manner, demonstrating the efficiency of our method and universality of the discovered reasoning structures.

**Applying GPT-4 Discovered Structures to Llama2 and ChatGPT**    Motivated by transferrability performance across LLMs, we further investigate can self-discovered reasoning structures from LLMs boost reasoning for *smaller LMs* that are challenging to come up with structures themselves[3]. We use GPT-4 to discover the task-intrinsic reasoning structures, and then apply those structures to the decoding of open-sourced Llama2-70B as well as GPT-3.5-turbo (ChatGPT) on two subsets of tasks from BBH. We find that using self-discovered structures on Llama2 (52%) outperforms CoT (42%) on disambiguation QA zero-shot and on GPT-3.5-turbo (56%) outperforms CoT (51%) on geometry with 3-shot demonstration from structured reasoning process.

## 6    Related Work

### 6.1    Prompting Methods

Recent advancements in the area of LLMs have given rise to a plethora of few-shot (Brown et al., 2020) and instruction (Mishra et al., 2022c; Wei et al., 2021; Ouyang et al., 2022) prompting techniques, including *Chain-of-Thought* prompting (CoT) (Nye et al., 2021; Wei et al., 2022), Least-to-most prompting (Zhou et al., 2022a; Drozdov et al., 2022), Decomposed prompting (Khot et al., 2022), Reframing (Mishra et al., 2022b), Help Me Think Prompting (Mishra & Nouri, 2023), Stepback Prompting (Zheng et al., 2023) and search-based approaches like *Tree-of-Thought (ToT)* (Yao et al., 2023a), Graph-of-Thought (Besta et al., 2023; Yao et al., 2023b), Branch-solve-merge (Saha et al., 2023) and RAP (Hao et al., 2023). Each of the prompting methods has some strengths and weaknesses in terms of their successful application domain. Our work SELF-DISCOVER presents the missing piece in the prompting literature, as SELF-DISCOVER provides a way to self-compose over various prompting methods via the proposed self-discovery mechanism. Composing over prompting methods in SELF-DISCOVER is analogous to the programming literature where a program is written using various basic building blocks such as for loop, if/else condition etc.

### 6.2    Reasoning and Planning

With the development of various reasoning and planning benchmarks such as GSM8K Cobbe et al. (2021), Math Hendrycks et al., BigBench Srivastava et al. (2023) etc., various methods have been proposed to improve model performance. Often these methods induce specific reasoning structures mimicking the reasoning structure of the underlying task associated with the dataset. For example, chain of thought Wei et al. (2022) and scratchpad Nye et al. (2021) induce generation of explanations associated with a reasoning question. Similarly other methods induces specific reasoning structures such as question summarization Kuznia et al. (2022), question decomposition Patel et al. (2022), program generation Mishra et al. (2022a); Chen et al. (2022); Gao et al. (2023b), etc. However, in a real world user traffic, queries can be diverse covering various reasoning structures. Our work SELF-DISCOVER allows models to combine multiple reasoning approaches by self-composing into a structure without the need to access task labels. There have been some related work that explores LLM combining skills in-context such as SkiC (Chen et al., 2023), devising a strategy (Gao et al., 2023a), and planning with iterative quering (Liu et al., 2023). However, they require human annotating skills and reasoning plans while SELF-DISCOVER leverages a scalable solution with the help of LLM's meta-task reasoning capabilities.

## 7    Conclusion

We introduce SELF-DISCOVER, an efficient and performant framework for models to self-discover a reasoning structure for any task from a seed set of general problem-solving skills. We observe drastic improvements on challenging reasoning benchmarks from multiple LLMs up to 30%. Ablations study of SELF-DISCOVER demonstrates that the composed reasoning structures are universally transferable between LLMs. Forward looking, we are excited to explore more on LLM structured reasoning to push the boundary of problem-solving and discover potentials for Human-AI collaboration.

---

[3]We tried zero-shot meta prompting Llama2 but observed low-quality structure outputs.

## Acknowledgement

We thank anonymous reviewers for their constructive feedback during the discussion period. We thank team members from Google DeepMind, INK Lab, and JAUNTS Lab for their insightful feedback on this paper. This work was funded in part by the Defense Advanced Research Projects Agency with awards HR00112220046, HR00112390061, and N660011924033.

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

# A    Self-Discover Prompt Details

Table 2 shows all 39 reasoning modules we use for SELF-DISCOVER, adopted from Fernando et al. (2023), that contain cognitive heuristics of problem-solving.

Figure 9 contains the structure of the three actions of SELF-DISCOVER during Stage 1, where it discovers an intrinsic reasoning structure on the task-level.

For Stage 2, where we use the self-discovered structure to solve the task instances, we start with the prompt: "*Follow the step-by-step reasoning plan in JSON to correctly solve the task. Fill in the values following the keys by reasoning specifically about the task given. Do not simply rephrase the keys.*", followed by the reasoning structure, and finally the task instance.

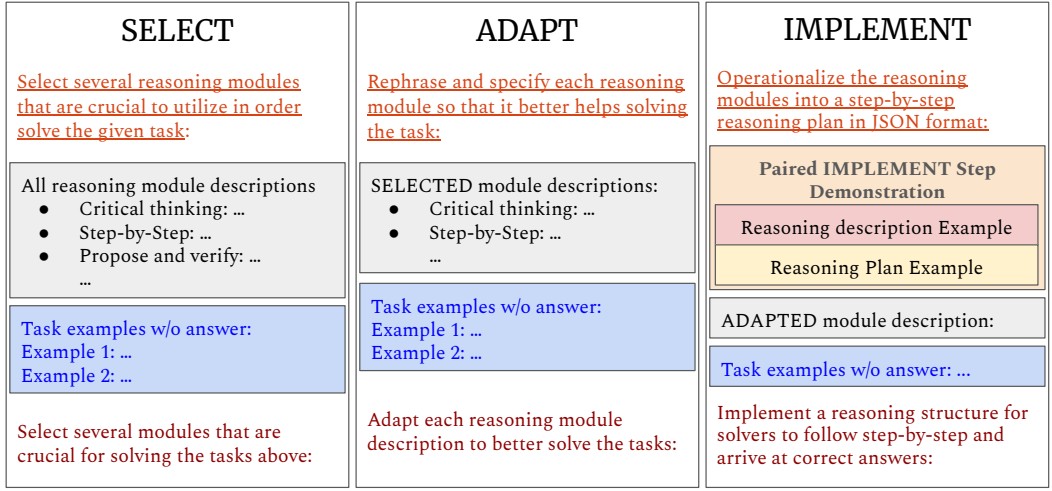

Figure 9: **Meta-Prompts for the three actions of SELF-DISCOVER**. Each meta-prompt consists of an instruction in the beginning and the end, reasoning module descriptions, and task examples without labels. For IMPLEMENT, to show model an example of a reasoning structure (plan), we present a human-written structure in JSON for another task.

# B    Evaluation Details

We use accuracy and exact matching as with other methods tested on BBH, T4D and MATH. To properly evaluate the generated answers from LLMs, we prompt the models to end the answer with "*Thus, the final answer is [X]*", where X is either one answer option such as "*A*" or a string such as "*valid*". During evaluation, we manually examine each task's outputs from LLMs and design heuristics to extract the final answers. For MATH dataset, we find that it is challenging to extract the answers accurately. As a result, we subsample 200 test examples from MATH, and manually sanity check and annotate the extracted answers for all methods tested in our paper.

Here we also provide more details on token counts for Self-Discover and other baselines. Since Self-Discover only added 3 calls per task, on per instance level it only needs to run once. The average length of self-discovered structures is 224 tokens for BBH, 183 tokens for T4D, and 152 for MATH, which is similar to ToT/GoT prompts with around 234 tokens. We run the stage 2 (generating solution based on the structure) only in 1 inference pass where the model fills the values in the self-discovered structures. Thus, Self-Discover still reduces much inference cost compared to CoT-Self-Consistency, majority voting, etc. which requires 20s-40s fold of each CoT reasoning path token counts.

# C    BBH Per Task Performance

Per-task performance on BBH (23 tasks in total) are shown in Table 3.

Table 2: All 39 reasoning modules consisting of high-level cognitive heuristics for problem-solving. We adopt them from Fernando et al. (2023).

**Reasoning Modules**

1 How could I devise an experiment to help solve that problem?
2 Make a list of ideas for solving this problem, and apply them one by one to the problem to see if any progress can be made.
3 How could I measure progress on this problem?
4 How can I simplify the problem so that it is easier to solve?
5 What are the key assumptions underlying this problem?
6 What are the potential risks and drawbacks of each solution?
7 What are the alternative perspectives or viewpoints on this problem?
8 What are the long-term implications of this problem and its solutions?
9 How can I break down this problem into smaller, more manageable parts?
10 Critical Thinking: This style involves analyzing the problem from different perspectives, questioning assumptions, and evaluating the evidence or information available. It focuses on logical reasoning, evidence-based decision-making, and identifying potential biases or flaws in thinking.
11 Try creative thinking, generate innovative and out-of-the-box ideas to solve the problem. Explore unconventional solutions, thinking beyond traditional boundaries, and encouraging imagination and originality.
12 Seek input and collaboration from others to solve the problem. Emphasize teamwork, open communication, and leveraging the diverse perspectives and expertise of a group to come up with effective solutions.
13 Use systems thinking: Consider the problem as part of a larger system and understanding the interconnectedness of various elements. Focuses on identifying the underlying causes, feedback loops, and interdependencies that influence the problem, and developing holistic solutions that address the system as a whole.
14 Use Risk Analysis: Evaluate potential risks, uncertainties, and tradeoffs associated with different solutions or approaches to a problem. Emphasize assessing the potential consequences and likelihood of success or failure, and making informed decisions based on a balanced analysis of risks and benefits.
15 Use Reflective Thinking: Step back from the problem, take the time for introspection and self-reflection. Examine personal biases, assumptions, and mental models that may influence problem-solving, and being open to learning from past experiences to improve future approaches.
16 What is the core issue or problem that needs to be addressed?
17 What are the underlying causes or factors contributing to the problem?
18 Are there any potential solutions or strategies that have been tried before? If yes, what were the outcomes and lessons learned?
19 What are the potential obstacles or challenges that might arise in solving this problem?
20 Are there any relevant data or information that can provide insights into the problem? If yes, what data sources are available, and how can they be analyzed?
21 Are there any stakeholders or individuals who are directly affected by the problem? What are their perspectives and needs?
22 What resources (financial, human, technological, etc.) are needed to tackle the problem effectively?
23 How can progress or success in solving the problem be measured or evaluated?
24 What indicators or metrics can be used?
25 Is the problem a technical or practical one that requires a specific expertise or skill set? Or is it more of a conceptual or theoretical problem?
26 Does the problem involve a physical constraint, such as limited resources, infrastructure, or space?
27 Is the problem related to human behavior, such as a social, cultural, or psychological issue?
28 Does the problem involve decision-making or planning, where choices need to be made under uncertainty or with competing objectives?
29 Is the problem an analytical one that requires data analysis, modeling, or optimization techniques?
30 Is the problem a design challenge that requires creative solutions and innovation?
31 Does the problem require addressing systemic or structural issues rather than just individual instances?
32 Is the problem time-sensitive or urgent, requiring immediate attention and action?
33 What kinds of solution typically are produced for this kind of problem specification?
34 Given the problem specification and the current best solution, have a guess about other possible solutions.
35 Let's imagine the current best solution is totally wrong, what other ways are there to think about the problem specification?
36 What is the best way to modify this current best solution, given what you know about these kinds of problem specification?
37 Ignoring the current best solution, create an entirely new solution to the problem.
38 Let's think step by step.
39 Let's make a step by step plan and implement it with good notion and explanation.

Table 3: Big Bench-Hard (Suzgun et al., 2022) per-task performance of GPT-4 and PaLM 2-L with SELF-DISCOVER.

| Big Bench-Hard Task | Human (Avg.) | Human (Max) | GPT-4 Direct | GPT-4 + CoT | GPT-4 + Self-Discover | PaLM 2-L Direct | PaLM 2-L + CoT | PaLM 2-L + Self-Discover |
|---|---|---|---|---|---|---|---|---|
| boolean_expressions | 79 | 100 | 73 | 83 | 85 | 71 | 84 | 84 |
| causal_judgement | 70 | 100 | 67 | 75 | 80 | 46 | 59 | 61 |
| date_understanding | 77 | 100 | 74 | 80 | 81 | 73 | 78 | 78 |
| disambiguation_qa | 67 | 93 | 60 | 70 | 80 | 54 | 50 | 57 |
| dyck_languages | 48 | 100 | 69 | 73 | 77 | 94 | 95 | 98 |
| formal_fallacies | 91 | 100 | 60 | 60 | 80 | 60 | 63 | 69 |
| geometric_shapes | 54 | 100 | 30 | 56 | 60 | 33 | 34 | 39 |
| hyperbaton | 75 | 100 | 68 | 69 | 76 | 80 | 75 | 82 |
| logical_deduction_seven_objects | 40 | 89 | 60 | 70 | 70 | 45 | 39 | 50 |
| movie_recommendation | 61 | 90 | 70 | 70 | 86 | 83 | 54 | 66 |
| multistep_arithmetic_two | 10 | 25 | 10 | 92 | 70 | 4 | 50 | 47 |
| navigate | 82 | 100 | 70 | 90 | 90 | 38 | 63 | 67 |
| object_counting | 86 | 100 | 90 | 100 | 100 | 27 | 44 | 70 |
| penguins_in_a_table | 78 | 100 | 80 | 100 | 90 | 70 | 67 | 75 |
| reasoning_about_colored_objects | 75 | 100 | 77 | 80 | 79 | 36 | 79 | 75 |
| ruin_names | 78 | 100 | 90 | 80 | 97 | 79 | 58 | 90 |
| salient_translation_error_detection | 37 | 80 | 40 | 50 | 70 | 56 | 48 | 60 |
| snarks | 77 | 100 | 73 | 89 | 97 | 58 | 62 | 86 |
| sports_understanding | 71 | 100 | 54 | 61 | 90 | 44 | 47 | 89 |
| temporal_sequences | 91 | 100 | 96 | 99 | 100 | 99 | 97 | 99 |
| tracking_shuffled_objects_seven_objects | 65 | 100 | 24 | 80 | 68 | 22 | 58 | 36 |
| web_of_lies | 81 | 100 | 15 | 80 | 71 | 54 | 42 | 67 |
| word_sorting | 63 | 100 | 65 | 90 | 85 | 12 | 4 | 15 |

```
reasoning_about_colored_objects          causal_judgement
{                                        {
    "Type and color of each item":          "Identify the chain of events in
    "Number of items of each color":     the story":
    "Number of items of each type":         "Identify the consequences of
    "Number of items of each color       each event":
and type":                                  "Identify the cause-and-effect
    "Final answer":                       relationships between events":
}                                            "Choose a final answer based
                                         on the reasoning":
                                         }
dyck_languages
{
        "Parentheses that are not closed properly":
        "Stack to store the closing parentheses":
        "If the next symbol is a closing parenthesis, pop the stack and
check if the popped symbol matches the next symbol":
        "If the stack is empty, add the next symbol to the stack":
}
```

*Break down to sub-tasks*

*Reflect on task nature*

*Devise an algorithm*

Figure 10: **Examples of self-discovered structures on BBH tasks using PaLM 2-L.** We observe traits of atomic reasoning modules such as "*step-by-step thinking*", "*reflect on task nature*", and an interesting creative thinking case where models devise an algorithm using *stack* to solve parenthesis parsing task.

Table 4: Additional baselines including Tree-of-Thought (ToT) and Graph-of-Thought (GoT) [Added rows].

| Method | BBH | T4D | MATH |
|---|---|---|---|
| PaLM 2-L | 56% | 30% | 45% |
| PaLM 2-L + CoT | 60% | 40% | 42% |
| PaLM 2-L + ToT | 58% | 41% | 44.5% |
| PaLM 2-L + GoT | 60% | 40% | 40% |
| PaLM 2-L + PS | 61% | 42% | 49% |
| PaLM 2-L + Self-Discover | **67%** | **69%** | **50.5%** |
| GPT-4 | 58% | 51% | 70.5% |
| GPT-4 + CoT | 75% | 52% | 71% |
| GPT-4 + ToT | 76% | 50% | 69% |
| GPT-4 + GoT | 75% | 52% | 70% |
| GPT-4 + PS | 73% | 53% | 70% |
| GPT-4 + Self-Discover | **81%** | **85%** | **73%** |

# D  Additional Experiments

We further include Tree-of-Thought (Yao et al., 2023a) and Graph-of-Thought (Besta et al., 2023) (zero-shot versions) as baselines for comparison shown in Table 4.

To show the effectiveness of Self-Discover on more general tasks, we tested on a subset of MMLU (10 subtasks, with 50 diverse questions each, all randomly sampled) and results are shown in Table 5. We find that GPT-4+Self-Discover wins GPT-4+CoT in zero-shot on 7 out of 10, ties on 2 out of 10, and loses on 1 out of 10 tasks. In addition, we tried Self-Discover on the instance-level, where for each question, we run stage 1 to output the reasoning structure, then solve the task. We find that the instance-level Self-Discover performs even better on MMLU, outperforming CoT by 7.2% on average for all tasks. This result, combined with those in main content, shows that the strength of Self-Discover spans across two types of tasks: for well-defined hard reasoning tasks such as BBH, task-level Self-Discover works well while being very efficient; for very open-domain tasks such as MMLU, we can do instance-level Self-Discover, which significantly outperforms CoT while still fewer inference required than self-consistency.

We show additional examples of self-discovered structures in Figure 10. We observe traits of atomic reasoning modules such as "*step-by-step thinking*", "*reflect on task nature*", and an interesting creative thinking case where models devise an algorithm using *stack* to solve parenthesis parsing task.

Table 5: MMLU (Suzgun et al., 2022) per-task performance of GPT-4 and PaLM 2-L with SELF-DISCOVER. We sampled 10 tasks with 50 examples each. SD (instance) refers to that we run stage one on each question and use the generated structure during solving, to acount for the diversity of questions. [New Table]

| MMLU Tasks | GPT-4 Direct | GPT-4 + CoT | GPT-4 +SD | GPT-4 +SD (instance) | PaLM 2-L Direct | PaLM 2-L + CoT | PaLM 2-L + SD | PaLM 2-L+SD (instance) |
|---|---|---|---|---|---|---|---|---|
| business_ethics | 78 | 83 | 85 | 91 | 72 | 77 | 80 | 83 |
| high_school_world_history | 64 | 69 | 74 | 83 | 54 | 59 | 61 | 66 |
| machine_learning | 72 | 80 | 81 | 88 | 70 | 75 | 75 | 78 |
| college_medicine | 45 | 52 | 50 | 54 | 44 | 45 | 45 | 49 |
| high_school_statistics | 68 | 75 | 75 | 84 | 60 | 66 | 68 | 73 |
| international_law | 70 | 77 | 77 | 82 | 60 | 69 | 63 | 71 |
| conceptual_physics | 62 | 66 | 70 | 74 | 59 | 64 | 65 | 69 |
| marketing | 71 | 75 | 76 | 82 | 67 | 69 | 71 | 74 |
| jurisprudence | 60 | 70 | 74 | 76 | 55 | 60 | 64 | 69 |
| moral_disputes | 62 | 68 | 69 | 73 | 60 | 65 | 66 | 68 |

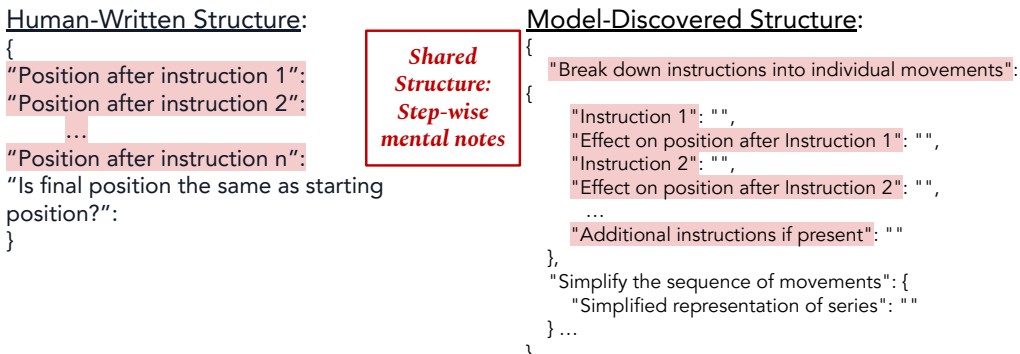

Figure 11: **Case study of human-written structure shares commonalities with LLM-discovered reasoning structure.** We observe similar reasoning patterns–both structures contain step-wise analysis of each instruction.

# E   Error Analysis

We perform an error analysis of SELF-DISCOVER on the MATH dataset of 200 samples to understand the failure modes. We manually annotate whether the generated reasoning structure is correct or not together with whether the correctness of model prediction using SELF-DISCOVER. A reasoning structure is defined as correct if a human expert can solve the task by simply following the reasoning structure.

Out of 200 examples, we find that 87.5% (175) examples have correct reasoning structures. 12.5% (25) examples have incorrect reasoning structures leading to prediction errors. Table 6 shows 4 such examples where the LLM misunderstands the task, or makes an error in one of the steps or adds unnecessary steps in the reasoning structure.

Next, we analyze the errors made by the model in SELF-DISCOVER: out of 99 examples where the model prediction is wrong, wrong reasoning structures account for only 25.3% of the errors. The remaining 74.7% errors are due to errors in the intermediate calculations such as math computations. Table 7 shows 3 examples of such errors. This insight indicates that future improvements should aim at improving the step-wise calculation accuracy of LLMs, such as using tools or code generation.

# F   Further Anaysis

**Model-Discovered Reasoning Structures vs. Human Reasoning Patterns**   We investigate whether LLM-discovered reasoning structures share some commonalities with human reasoning patterns. We give humans 3 task instances without labels and an example reasoning structure (same as SELF-

Table 6: Examples of wrong reasoning structures for MATH. The first error in the reasoning structure is highlighted in red.

| Prompt | Reasoning Structure | Error |
|---|---|---|
| How many numbers between 1 and 2005 are integer multiples of 3 or 4 but not 12? | 1. Find the number of multiples of 3 between 1 and 2005.
2. Find the number of multiples of 4 between 1 and 2005.
3. Find the number of multiples of 12 between 1 and 2005.
4. Add the number of multiples of 3 and 4.
5. Subtract the number of multiples of 12 (to avoid double counting).
6. The result is the number of integers between 1 and 2005 that are multiples of 3 or 4 but not 12. | Need to subtract the number of multiples of 12 twice instead of once. |
| How many numbers are in the list 6,7,10,11,14,15,..., 94,95,98? | 1. Find the number of pairs in the list.
2. Find the number of numbers in each pair.
3. Multiply the number of pairs by the number of numbers in each pair to find the total number of numbers in the list. | LLM misunderstands the task: the reasoning structure is irrelevant. |
| A restaurant offers three desserts, and exactly twice as many appetizers as main courses. A dinner consists of an appetizer, a main course, and a dessert. What is the least number of main courses that the restaurant should offer so that a customer could have a different dinner each night in the year 2003? | 1. Define the variables:
- M: number of main courses
- A: number of appetizers (twice the number of main courses)
- D: number of desserts (fixed at 3)
- T: total number of possible dinner combinations
2. Calculate the total number of possible dinner combinations:
- T = M * A * D
3. Set the total number of possible dinner combinations equal to the number of nights in the year 2003:
- T = 2003
4. Substitute the expression for A in terms of M:
- T = M * (2 * M) * 3
5. Solve the equation for M:
- $6M^2 = 2003$
6. Find the smallest integer value of M that satisfies the equation. | There are 365 nights in the year 2003. |
| Determine the number of ways to arrange the letters of the word NINE. | 1. Count the total number of letters in the word NINE.
2. Identify the number of times each letter appears in the word.
3. Calculate the factorial of the total number of letters.
4. Divide the factorial by the factorial of the number of times each letter appears.
5. Multiply the results from step 4 for each unique letter. } | Step 5 is unnecessary and leads to error in the final result. |

Table 7: Examples of wrong calculations for MATH. The first error in the intermediate computations is highlighted in red.

| Prompt | Self-Discover Prediction | Error |
|---|---|---|
| How many ways are there to arrange 6 people around a circular table with 7 seats? (Two seatings are considered the same if one is a rotation of the other.) | "1. Calculate the total number of ways to arrange 6 people in 7 seats without considering rotations as the same.": 7 * 6 * 5 * 4 * 3 * 2 = 5040
- There are 7 choices for the first person.
- There are 6 choices for the second person.
- There are 5 choices for the third person.
- There are 4 choices for the fourth person.
- There are 3 choices for the fifth person.
- There are 2 choices for the sixth person.
"2. Divide the result from step 1 by the number of rotations that are considered the same.": 5040 / 6 = 840
- There are 6 rotations that are considered the same. | There are 7 rotations instead of 6 that are considered the same. |
| A customer ordered 15 pieces of gourmet chocolate. The order can be packaged in small boxes that contain 1, 2 or 4 pieces of chocolate. Any box that is used must be full. How many different combinations of boxes can be used for the customer's 15 chocolate pieces? One such combination to be included is to use seven 2-piece boxes and one 1-piece box. | "1. Find the number of ways to package the chocolates using only boxes of 1 piece.": 1,
"2. Find the number of ways to package the chocolates using only boxes of 2 pieces.": 1,
"3. Find the number of ways to package the chocolates using only boxes of 4 pieces.": 0,
"4. Find the number of ways to package the chocolates using a combination of boxes of 1 and 2 pieces.": 7,
"5. Find the number of ways to package the chocolates using a combination of boxes of 1 and 4 pieces.": 3,
"6. Find the number of ways to package the chocolates using a combination of boxes of 2 and 4 pieces.": 2,
"7. Find the number of ways to package the chocolates using a combination of boxes of 1, 2, and 4 pieces.": 4,
"8. Sum up the total number of ways from the seven cases above.": 1 + 1 + 0 + 7 + 3 + 2 + 4 = 18 | There is 0 way to package 15 chocolates using only boxes of 2 pieces. |
| How many primes are in the row of Pascal's Triangle that starts with a 1 followed by a 6? | "1. Identify the row of Pascal's Triangle that starts with a 1 followed by a 6.": "1, 6, 15, 20, 15, 6, 1",
"2. List all the numbers in that row.": [1, 6, 15, 20, 15, 6, 1],
"3. Check each number in the list for primality.": [False, True, False, False, False, True, False],
"4. Count the number of primes found in the list.": 2 | 6 is not a prime. |

DISCOVER meta-reasoning stage) and ask them to write a reasoning structure for a task before solving it. Figure 11 shows comparison of human and LLM-composed reasoning structures on the BBH-navigation task. We observe similar structures such as mental-noting after each movement. From promising findings of LLM self-discovered structures boost and share traits of human meta-reasoning, we hope to encourage more future work to study humna-AI collaboration for complex problem-solving.

To show whether the chosen reasoning modules and discovered structures provide a diverse coverage, we have included more details on frequency of selected reasoning modules in Table 8. Furthermore, we observe a very diverse set of self-discovered structures as the 12 examples shown in Figure 6, 10, 11, and Table 6 and 7. Math problem structures (Table 6) tend to be very different from BBH ones (Figure 10). Due to the non-deterministic nature of LLMs, the number of reasoning structures that can be discovered by Self-Discover is infinite, because it can use multiple reasoning modules in many different orders. In our prompts, we specifically do not restrict how the structures should use the modules and find examples where models use new modules not in the seed list (Figure 10 Dyck-Language example where model devises a stack algorithm, which is not in the seed list).

Table 8: Selected reasoning module frequency on 25 sub tasks (BBH, T4D and MATH), for full reasoning module descriptions, please refer to Table 2 in Appendix. Top 5 selected modules are in bold.

| Reasoning Module Index | Short Module Description (For Full set see Table 2) | Frequency Over 25 Tasks |
|---|---|---|
| 1 | Devise an experiment | 9/25 |
| 2 | Make a list of ideas | 4/25 |
| 3 | Measure Progress | 6/25 |
| 4 | **Simplify the problem** | **17/25** |
| 5 | Key assumptions | 10/25 |
| 6 | Risks of solution | 3/25 |
| 7 | Alternative perspectives | 12/25 |
| 8 | Long-term implications | 2/25 |
| 9 | Break down to smaller problems | 14/25 |
| 10 | **Critical thinking** | **18/25** |
| 11 | Creative thinking | 12/25 |
| 12 | Seek input from others | 5/25 |
| 13 | Systems thinking | 11/25 |
| 14 | Risk analysis | 3/25 |
| 15 | Reflective thinking | 13/25 |
| 16 | Core issues? | 7/25 |
| 17 | Underlying causes or factors | 6/25 |
| 18 | Solutions tried before | 2/25 |
| 19 | Obstacles or challenges? | 10/25 |
| 20 | Relevant data? | 9/25 |
| 21 | Stakeholder needs? | 1/25 |
| 22 | Resources needed? | 8/25 |
| 23 | Evaluate progress? | 9/25 |
| 24 | Metrics can be used? | 7/25 |
| 25 | Technical or practical problem | 4/25 |
| 26 | Physical constraint? | 1/25 |
| 27 | Human behavior? | 3/25 |
| 28 | **Decision-making or planning** | **15/25** |
| 29 | Analytical or optimization | 7/25 |
| 30 | Design challenge? | 2/25 |
| 31 | Systemic or structural issues? | 4/25 |
| 32 | Time-sensitive? | 1/25 |
| 33 | Typical solutions? | 5/25 |
| 34 | Other possible solutions | 12/25 |
| 35 | Imagine current solution is wrong | 8/25 |
| 36 | Modify current solution | 7/25 |
| 37 | Create entirely new solution | 5/25 |
| 38 | **Think step-by-step** | **17/25** |
| 39 | **Step-by-step plan with explanation** | **21/25** |

