# OpenReview forum: "SELF-DISCOVER: Large Language Models Self-Compose Reasoning Structures"
_NeurIPS.cc/2024/Conference — NeurIPS 2024 poster_

### Official Review · Reviewer_NVyo · 2024-06-26

**Soundness:** 3
**Presentation:** 3
**Contribution:** 3
**Rating:** 6
**Confidence:** 4

**Summary:**

The paper introduces a new prompting method, Self-Discover, that enhances the complex reasoning abilities of Large Language Models. Self-Discover consists of two stages. At the first one, the method develops a task-adapted prompt by performing 3 steps: SELECT, when the LLM selects the reasoning modules (RMs) from a pre-defined list, ADAPT, when the model adapts the chosen reasoning modules to the task, and IMPLEMENT, when the model, conditioned on the adapted reasoning modules, generates the reasoning structure as a JSON. Each step is performed in a zero-shot manner and controlled by a dedicated meta-prompt. The resulting reasoning structure is used in the final second stage, as part of the prompt to solve each task and produce the output. For each task instance, the LLM first reasons by completing the JSON value fields, and then outputs the answer. Self-Discover is compared to several baselines: Direct Prompting, CoT (+Self-Consistency), Plan-and-Solve, Majority voting of RMs, and Best of each RM, as well as Graph-of-Thought (GoT) and Tree-of-Thought (ToT). Evaluation is performed on BIG-Bench Hard (BBH), Thinking for Doing (T4D), and (subsampled) MATH benchmarks. The results demonstrate the superiority of Self-Discover against the baselines above in terms of performance while maintaining comparatively low computational cost, expressed in the number of necessary inference calls. The ablation study highlights that each step of Stage 1 significantly contributes to the overall performance. The generated reasoning structures appear to be general and transferable between different LLMs, compared to Optimized Prompts (OPRO).

**Strengths:**

- The paper proposes a simple yet effective approach that greatly improves the reasoning abilities of LLMs. This is especially emphasized by the low computational costs of the method of only 3 extra inference calls per task. The method unifies multiple prompting-reasoning techniques and thus is scalable and adaptable.
- The evaluations are purposeful, and every major claim of the paper is supported by empirical evidence, including all the necessary ablation studies justifying the steps of Stage 1. A thorough analysis of the methods' performance, showing tasks where the method excels more, is included in the paper.

**Weaknesses:**

- The paper doesn't emphasize enough that the method requires task adaptation. This is a crucial detail, as, unlike CoT or ToT, Self-Discover cannot be deployed as a general zero-shot agent, but requires some specialized prompting with task examples. This limitation is addressed barely in the paper. Though this is not a huge disadvantage, it places the method right between zero-shot methods (CoT / ToT / GoT / PS) and task-adaptive methods (OPRO / DSPy).  Thus, more comparisons with the prompt adaptation techniques should be present in the paper, and the efficiency of Self-Discover should be emphasized when comparing to them.
- The authors claim that their method "composes atomic reasoning models into a reasoning structure", but this seems misleading. Figure 4 demonstrates that the method only slightly outperforms the best RM baseline, suggesting that no actual "composition" is happening, but only a selection and adaptation of the best fitting reasoning technique. This requires further investigation, for example, by analyzing which reasoning modules were picked for a task, and ablating the performance on these modules. This would reveal whether the model indeed uses multiple reasoning modules or only one in its prediction.

**Questions:**

- How robust is the method to changes in the list of reasoning modules? An ablation study on them, or at least the list of chosen modules for each task would reveal this information. Also, how robust is the method to the set of chosen task examples? What was their number in the experiments, and does varying their number or themselves affect the performance?
- What were the parameters of ToT & GoT in the experiments? The value of the breadth parameter can largely affect the performance of these methods.
- What modifications can be applied to the set of RM to increase the performance on the algorithmic tasks? I assume some specialized algorithmic prompting can be useful. What you also include the relationship between reasoning modules and the tasks in which they are useful, and assert that the model chooses the correct reasoning module?

**Limitations:**

As mentioned above, though the method is very effective, it is dependent on the set of chosen task examples, and so is task-adaptable, and not completely zero-shot. This limitation is not emphasized enough in the paper.

However, overall, it is a nice paper with promising results, which once again underscores the power of optimal prompting and alignment.

---

> ### Author Rebuttal · Authors · 2024-08-07
>
> Thank you so much for your helpful feedback and suggestions. Our responses to weaknesses and questions are as follows:
>
> __[Task adaptation]__
>
> Thank you for the great point. We’d like to first clarify that though a few task examples are used in meta-prompting, Self-Discover does NOT require any answer to the examples. Thus, OPRO and other task-adaptation methods such as StrategyLLM are not directly comparable to Self-Discover because ORPO is a supervised prompting tuning method needing 100+ training examples whereas Self-Discover does not need any answer labels, similarly to StrategyLLM, which requires accuracy on a validation set for strategy optimization. We will make the task-adaptation clearer in the paper.
>
> __[Reasoning module analysis and Q1]__
>
> Here we provide more details on the reasoning module selection process. First, we observe that on harder reasoning tasks (MATH and some subtasks from BBH), best of a single RM lags behind Self-Discover by a larger margin.Second, we did find that several reasoning modules (critical thinking, step-by-step, simplifying) are picked up more frequently than others and we’ve attached the detailed frequency in the new 1-page pdf. Third, we conducted an ablation study on the chosen reasoning modules on BBH and observed an average of 4.7% drop in performance with an average minimum drop of 2.5%. This indicates that multiple reasoning modules work synergistically for structured reasoning. We plan to include more comprehensive ablation studies on all the tasks for every reasoning module using a newer cheaper version of GPT-4.
>
> __[Q1: Robustness to task examples]__
>
> We randomly sampled 5 times of task examples and observed +-1% in performance difference, showing that the method is not impacted by task example choices.
>
> __[Q2: ToT and GoT parameters]__
>
> To keep comparable with Self-Discover, we use the zero-shot version of ToT and GoT and prompt models to conduct reasoning with a breadth of 5. We tested between 3 to 7 and did not observe significant differences.
>
> __[Q3: Modifications on RM]__
>
> Generally, we think first expanding the current set by prompting models and then pruning the generated set to remove redundancy would be a good starting point. Specifically, if we can use a specialized RM for say MATH and coding problems, we imagine it would drastically improve the reasoning performance!

---

> > ### Comment · Reviewer_NVyo · 2024-08-13
> >
> > Thanks to the authors for their response.
> >
> > My concerns regarding the synergy between the reasoning modules have been addressed. However, as the authors mentioned, I expect to see more ablation results in the appendix in the paper revision, which is why I intend to keep the positive score.

---

> > > ### Author Response · Authors · 2024-08-14
> > >
> > > We are glad that your concerns were addressed and we will update in the appendix more ablations in the next revision. We thank the reviewer for helping us to improve our paper!

---

### Official Review · Reviewer_nKMJ · 2024-06-30

**Soundness:** 3
**Presentation:** 3
**Contribution:** 2
**Rating:** 4
**Confidence:** 4

**Summary:**

The paper introduces SELF-DISCOVER, a framework enabling large language models (LLMs) to autonomously identify and utilize intrinsic reasoning structures to tackle various reasoning tasks. The framework is applicable across different model families and aligns with human reasoning patterns.

**Strengths:**

1. The paper introduces a novel framework, SELF-DISCOVER, that allows LLMs to autonomously identify and use task-specific reasoning structures.
2. The method is effective across various benchmarks.

**Weaknesses:**

1. The framework's effectiveness is heavily dependent on the quality and comprehensiveness of the atomic reasoning modules available, which may require significant human effort to define. Additionally, there is no guideline provided on how to develop these modules.
2. The main results are reported using a very limited range of models, specifically GPT-4 and PaLM-2. A systematic evaluation across a broader range of models, such as GPT-3.5, Llama2, and Llama3, is necessary, as merely applying reasoning structures generated by GPT-4 to these models is insufficient.
3. The compared baselines, such as zero-shot CoT and Plan-and-Solve, are not strong enough. Automatic prompting engineering methods, like LLM as Optimizers [1] and StrategyLLM [2], should be included in comparisons and discussions.

[1] Large Language Models as Optimizers. https://arxiv.org/abs/2309.03409

[2] StrategyLLM: Large Language Models as Strategy Generators, Executors, Optimizers, and Evaluators for Problem Solving. https://arxiv.org/abs/2311.08803

**Questions:**

NA

**Limitations:**

Yes

---

> ### Author Rebuttal · Authors · 2024-08-07
>
> Thank you so much for your helpful feedback and suggestions. Our responses to weaknesses and questions are as follows:
>
> __[Dependent on quality of atomic reasoning modules]__
>
> We would like to point out that the main contribution of this paper is the Self-Discover process. To show its effectiveness despite simple and not human-crafted seed reasoning modules, we took modules available from the Promptbreeder paper [1] and showed that it worked. Future work can certainly expand to more prompts to increase even more on the diversity and effectiveness of the method.
>
> __[Range of models]__
>
> In Section 5.2, we specifically tested GPT-3.5 and Llama2’s effectiveness on Self-Discover reasoning structures and found it outperformed CoT by a large margin. As footnote 3 suggested, we initially tried using Llama2 and GPT-3.5 to zero-shot self-discover reasoning structures (Llama3 was not made available at the time of submission) but observed low-quality structure outputs. Meta-reasoning capabilities seem to require stronger base LLMs. We believe that more LLMs will become increasingly capable of performing self-discover reasoning and more accessible.
>
> __[Compared baselines]__
>
> In addition to CoT and Plan-and-Solve, we also compared Tree-of-Thought and Graph-of-Thought and observed similar improvements (Appendix Table 4). We will put these result highlights in the main content. Furthermore, we compared LLM as Optimizers (OPRO) on transferability in Figure 8. Note that OPRO and StrategyLLM is not directly comparable to Self-Discover because ORPO is a supervised prompting tuning method with 100+ examples whereas Self-Discover is zero-shot, similarly to StrategyLLM (cited in paper on line 314) which requires accuracy on a validation set for strategy optimization.
>
> [1] Fernando, Chrisantha, et al. "Promptbreeder: Self-referential self-improvement via prompt evolution." arXiv preprint arXiv:2309.16797 (2023).

---

> > ### Author Response · Authors · 2024-08-13
> >
> > Thanks again for reviewing our work! Please let us know if we have addressed your questions and concerns.

---

### Official Review · Reviewer_Kd8z · 2024-07-08

**Soundness:** 3
**Presentation:** 3
**Contribution:** 2
**Rating:** 5
**Confidence:** 5

**Summary:**

This paper introduces SELF-DISCOVER, a framework designed to enhance the reasoning capabilities of Large Language Models (LLMs) such as GPT-4 and PaLM 2. The framework enables LLMs to self-discover and compose intrinsic reasoning structures tailored to specific tasks, improving performance on complex reasoning benchmarks.

**Strengths:**

- By enabling LLMs to self-compose reasoning structures, this method significantly improves performance on complex reasoning tasks, demonstrating the ability to handle intricate problems more effectively than traditional methods.

- SELF-DISCOVER achieves superior results while requiring substantially fewer inference steps (10-40x fewer) compared to inference-intensive approaches like CoT-Self-Consistency.

- The reasoning structures discovered by the SELF-DISCOVER method show strong transferability across different LLM families, highlighting its broad applicability and robustness.

**Weaknesses:**

1. While atomic reasoning structures form the foundation of the SELF-DISCOVER method, their inherent simplicity can restrict the overall reasoning performance. These basic units, although useful, may not fully capture the complexity needed for certain advanced reasoning tasks, potentially limiting the depth and flexibility of the reasoning process.

2. The proposed method is significantly reliant on the performance of the underlying LLM. Factors such as the model's ability to follow instructions accurately and the limitations imposed by context window sizes can pose significant challenges. For instance, if the base LLM struggles with instruction-following or is constrained by a limited context window, the effectiveness of the SELF-DISCOVER method may be compromised, limiting its potential for broader application and scalability.

3. The iterative process of selecting, adapting, and implementing reasoning modules introduces extra computational costs. Each step in this process requires additional queries to the LLM, which can accumulate and result in increased computational demands. This overhead may offset some of the efficiency gains achieved through reduced inference steps, particularly in scenarios requiring frequent adaptation and customization of reasoning structures.

**Questions:**

1. What is the additional cost of implementing this method, such as the extra tokens generated or the increase in inference time?

2. Please refer to the correct section number in the Appendix in lines 83-84.

3. The comparison in Figure 5 may not be fair. The self-discovery method requires fewer calls but could lead to more token generations.

4. The design of the "select, adapt, and implement" steps seems ad-hoc. Why are these three steps proposed separately? For instance, why can't "select" and "adapt" be combined into a single step since their prompts are similar, as shown in Figure 9?

5. What are the commonly selected reasoning structures? Are all 39 reasoning modules useful? It would be better to show the density of the selected reasoning modules to ensure that the selection of 39 modules is meaningful.

6. Does the model perform consistently in the selection process?

7. Do different models, such as GPT-4 and Palm-2, perform consistently? Do they discover similar reasoning structures?

**Limitations:**

This paper does not sufficiently illustrate the limitations of the proposed method. The checklist on lines 542-543 states, "We present an extensive analysis of the limitations of Self-Discover in both Sections 4 and 5 with an extended error analysis on MATH in Appendix E." However, Sections 4 and 5 are primarily general evaluations of the proposed method and do not contain sufficient and clear illustrations of its limitations. This issue needs to be strictly addressed.

---

> ### Author Rebuttal · Authors · 2024-08-07
>
> Thank you so much for your helpful feedback and suggestions. Our responses to weaknesses and questions are as follows:
>
> __[Simplicity of atomic reasoning modules]__
>
> Thank you for the great point and we agree that the 39 atomic reasoning modules can be improved. We would like to note two points: 1. This paper’s main message is to show the effectiveness of the self-discover method despite the inherent simplicity of the atomic reasoning structures used here. 2. Future work can grow and expand the atomic reasoning structure to increase their diversity, complexity to capture more advanced reasoning tasks.
>
> __[Reliant on underlying LLM]__
>
> We believe that LLMs will become increasingly capable for instruction following and with longer context windows, even with smaller models. For example post-training for instruction-following is considered basic requirements for any LLMs to work. And the context window size is typical for the community standard of 2K tokens. Therefore, we think that the required capabilities in LLMs for Self-Discover to work are common for current LLMs and will become increasingly accessible (such as Llama 3).
>
> __[Computation overhead and Q1, 3]__
>
> Given a task, Self-Discover only adds 3 more inference calls on the task level (select, adapt, and implement), then 1 inference call per instance of the task (lines 84-87). The average token # for the task-level 3-step prompting is around 1200 (select-input), 212 (select-output), 230 (adapt-input), 250 (adapt-output), 291 (implement-input), and 193 (implement-output) and for instance-level prompting is around 212 (input) and 265 (output). We would like to emphasize that the higher token cost is per-task and thus the amortized cost per-instance is less than baselines that use iterative inference methods such as CoT-Self-Consistency and fewer token length as Tree/Graph of Thought.
>
> __[Questions]__
>
> Q2: We will fix the section number in the next version.
>
> Q4: 3-step design: Initially we tried fewer steps in the first stage, but found that the final reasoning structure is often low quality (also supported in ablation studies Figure 7). Then we split the discovery phase into 3 steps (SELECT, ADAPT, IMPLEMENT) so the difficulty of generating a structure directly can be mitigated, and find that the quality much improved. Another reason we separate SELECT and ADAPT is that in the meta prompt: we give LLMs freedom to make changes of obviously wrong selected/adapted modules by instructing “Feel free to change or come up with new reasoning modules” and find this improved discovered structure quality.
>
> Q5: We have included more details on frequency of selected reasoning modules in the attached 1-page pdf. Common modules include critical thinking, step-by-step reasoning, simplifying, etc. Note that even though the same module might be selected in different tasks, the final reasoning structure might look very different and customized to the task itself.
>
> Q6 and 7: Yes, models perform consistently in the reasoning module selection process across a wide range of tasks. We observe that PaLM-2 and GPT-4 result in similar structures on BBH, but vary more on MATH tasks, indicating potentially different mathematical reasoning post-training procedures.

---

> > ### Comment · Reviewer_Kd8z · 2024-08-12
> >
> > I thank authors for the response, please see my comments below.
> >
> > > We have included more details on frequency of selected reasoning modules in the attached 1-page pdf.
> >
> > Please add this table in the appendix in the paper revision.
> >
> > >  models perform consistently in the reasoning module selection process across a wide range of tasks.
> >
> > Please also include these details in the paper revision.
> >
> > Based on the above discussions, I will keep my positive score.

---

> > > ### Author Response · Authors · 2024-08-14
> > >
> > > We thank the reviewer for the feedback and we will include the additional details in the appendix in the next revision.

---

### Official Review · Reviewer_i8mj · 2024-07-11

**Soundness:** 3
**Presentation:** 3
**Contribution:** 2
**Rating:** 4
**Confidence:** 4

**Summary:**

The paper introduces SELF-DISCOVER, a novel framework that enables large language models (LLMs) to self-discover and compose reasoning structures for tackling complex reasoning tasks. The core of SELF-DISCOVER is a self-discovery process where LLMs select multiple atomic reasoning modules, such as critical thinking and step-by-step thinking, and compose them into an explicit reasoning structure to follow during decoding.

**Strengths:**

1) SELF-DISCOVER significantly improves the performance of state-of-the-art LLMs on complex reasoning tasks, showcasing substantial gains over traditional prompting methods like Chain of Thought (CoT).
2) The discovered reasoning structures are not only effective but also transferable across different model families, indicating a level of universality in the approach. SELF-DISCOVER provides a more interpretable way to understand the reasoning process of LLMs by generating explicit reasoning structures.
3)  The paper includes a thorough empirical evaluation on a diverse set of reasoning tasks, demonstrating the framework's effectiveness across various domains.

**Weaknesses:**

1) The paper does not specify the exact number of reasoning structures that can be discovered by the SELF-DISCOVER framework. It's unclear whether the framework can generate a wide variety of structures or if it tends to converge on a limited set of common structures.
2) It is not detailed whether different examples within the same task utilize the same reasoning structure or if the framework can adapt the structure to suit the nuances of individual examples.
3) The paper's comparative analysis may be considered basic, as it does not delve into a comprehensive set of existing methods or include recent state-of-the-art approaches for comparison.
4) Although the paper mentions a similarity to the "Meta Reasoning for Large Language Models" approach, it does not provide a direct comparative experiment or analysis to highlight differences and potential advantages of SELF-DISCOVER.
5) The paper might not fully account for or cite the most recent literature, specifically works from 2024 and beyond, which could provide additional context and comparison points for the SELF-DISCOVER framework.

**Questions:**

See weaknesses.

**Limitations:**

See weaknesses.

---

> ### Author Rebuttal · Authors · 2024-08-07
>
> Thank you so much for your helpful feedback and suggestions. Our responses to weaknesses and questions are as follows:
>
> __[Variety of structures]__
>
> We have included more details on frequency of selected reasoning modules in the attached 1-page pdf. Furthermore, we observe a very diverse set of self-discovered structures as the 12 examples shown in Figure 6, 10, 11, and Table 6 and 7. Math problem structures (Table 6)  tend to be very different from BBH ones (Figure 10). Due to the non-deterministic nature of LLMs, the number of reasoning structures that can be discovered by Self-Discover is infinite, because it can use multiple reasoning modules in many different orders. In our prompts, we specifically do not restrict how the structures should use the modules and find examples where models use new modules not in the seed list (Figure 10 Dyck-Language example where model devises a stack algorithm, which is not in the seed list).
>
> __[Examples within the same task use the same structure]__
>
> Self-Discover Stage 1 (generate a structure from the task) operates on task-level (Figure 2), thus each example receives the same structure. This ensures the efficiency of the Self-Discover approach because otherwise, we have to run multiple inferences per instance of task (shown in Figure 5). Even if we do not tailor to each example of each task, we observe significant performance increases compared to CoT, CoT, GoT, Plan-and-Solve (Table 4). Future work can investigate how we can tailor the reasoning structure to each example efficiently.
>
> __[Comparative analysis]__
>
> We compared many SoTA methods including CoT, Plan-and-Solve, Self-Consistency, OPRO, ToT and GoT (Tables 1 and 4, Figures 5 and 8).
>
> __[Differences from other meta reasoning]__
>
> In Section 6.2, we discussed differences with related work of meta reasoning. We didn’t directly compare with methods such as SkiC, strategy reasoning, and iterative planning because they require human efforts (annotating skills and reasoning plans) or training examples (to tune prompt or optimize strategy) while SELF-DISCOVER focuses on proposing a scalable solution with the help of LLM’s meta-task reasoning capabilities. We did include Plan-and-Solve, which is directly comparable, in our results and find Self-Discover significantly outperformed it (Table 1 and Figure 5)
>
> __[2024 and beyond work]__
>
> Thank you for the note! We included related work to the best of our knowledge. We will definitely include the most recent work from 2024 and beyond if you can kindly provide us with pointers.

---

> > ### Author Response · Authors · 2024-08-13
> >
> > Thanks again for reviewing our work! Please let us know if we have addressed your questions and concerns.

---

### Official Review · Reviewer_5mv1 · 2024-07-16

**Soundness:** 3
**Presentation:** 3
**Contribution:** 2
**Rating:** 6
**Confidence:** 4

**Summary:**

This paper proposes a prompt engineering scheme:
Given a task and 39 prompts for solving tasks,
an LLM is prompted to select which of the 39 prompts are suitable for the task.
The selected prompts are then rephrased to be more specific to the task,
and reformatted in json.

The paper uses GPT-4 Turbo, PaLM 2-L, and Llama2-70B
and compares direct prompting, chain of thought, plan and solve,
chain of though with self-consistency, majority vote of the 39 prompts, and the proposed method.

**Strengths:**

1. This prompting scheme performs best on tasks that require diversity,
and the work demonstrates the value of prompt diversity.

2. Ablation experiments show the increased accuracy for each of the three steps of selecting prompts, rephrasing them, and reformatting (or implementation), over chain of thought.

**Weaknesses:**

1. As a prompt engineering method, it is unclear if the choice of the specific 39 prompts will withstand the test of time across LLMs and tasks.

2. Recent prompt engineering tools, such as Anthropic's Claude 3.5 Prompt Engineer, receive a task as input and output a detailed prompt best suited for the task and LLM, which may supersede this approach.

3. The claim of efficiency should be more rigorously validated and consider both the number of calls and the token lengths of the prompts and responses of each call.

**Questions:**

When comparing prompting schemes, it is important to give equal compute time to different methods.
When measuring efficiency did the work consider the lengths of the prompts and responses?

**Limitations:**

Yes

---

> ### Author Rebuttal · Authors · 2024-08-07
>
> Thank you so much for your helpful feedback and suggestions. Our responses to weaknesses and questions are as follows:
>
> __[Specific 39 seed modules]__
>
> Thanks for the very important point. The main contribution of the paper is the self-discover method and we demonstrate its effectiveness with the current 39 reasoning modules. Future work can certainly expand to more prompts to increase diversity and effectiveness of the method.
>
> __[Comparison to Anthropic’s PromptEngineer]__
>
> As a piece of science work, we fully and openly discuss how our method discovers a structured prompt using LLM meta prompts with details. It is unclear how Prompt Engineer from Claude works as it is proprietary. We believe papers such as Self-Discover present an important step to open science.
>
> __[Efficiency: number of calls and token lengths]__
>
> We compared the number of calls per instance in Figure 5 x-axis. Since Self-Discover only added 3 calls per task, on per instance level it only needs to run once. The average length of self-discovered structures is 224 tokens for BBH, 183 tokens for T4D, and 152 for MATH, which is similar to ToT/GoT prompts with around 234 tokens. We run the stage 2 (generating solution based on the structure) only in 1 inference pass where the model fills the values in the self-discovered structures. Thus, Self-Discover still reduces much inference cost compared to CoT-Self-Consistency, majority voting, etc. which requires 20s-40s fold of each CoT reasoning path token counts. We will add such details in the next version.

---

### Author Rebuttal · Authors · 2024-08-07

We thank all the reviewers for their time and insightful comments. We are pleased to receive this positive feedback from reviewers, particularly:

- Significantly improves performance across diverse complex reasoning tasks and demonstrates the value of prompt diversity (Reviewer i8mj, nKMJ, and 5mv1)
- Simple yet effective approach that greatly improves the reasoning abilities of LLMs (Reviewer NVyo)
- The discovered reasoning structures indicate a level of transferability/universality, highlighting its broad applicability and robustness (Reviewer i8mj and Kd8z)
- Self-Discover provides an interpretable way to understand the reasoning process of LLMs (Reviewer i8mj)
- The evaluations are purposeful, and every major claim of the paper is supported by empirical evidence with ablations (Reviewer 5mv1 and NVyo)


In addition to the above comments, we received valuable feedback from the reviewers, and included a 1-page pdf containing detailed selected reasoning module statistics for our tasks addressing common reviewer questions.

We have addressed most of the feedback in the comments and will improve the paper in the final version. We thank you again for all the reviewers' efforts to make our paper better and please let us know if you have any more questions.

---

### Decision · Program_Chairs · 2024-09-25

**Decision:**

Accept (poster)

**Comment:**

Among an already dense field of prompting approaches, the paper presents a compelling and novel approach to prompting based on self-discovery of a reasoning structure and then adaptation of this structure to the specific task at hand.  The idea is inspired by studies of human problem solving that have also motivated other prompting methodologies.  Extensive (and as reviewer NVyo points out -- purposeful) experimental results not only show the improvement of this method over numerous baselines, but also extensive ablations, quantitative, qualitative investigation of its performance.  The paper is well-written.  Rebuttal discussion led to analysis updates, clarifications, and agreement to add additional analysis in the Appendix, which the authors are requested to include in their final revision.